# Kinase Inhibition in Relapsed/Refractory Leukemia and Lymphoma Settings: Recent Prospects into Clinical Investigations

**DOI:** 10.3390/pharmaceutics13101604

**Published:** 2021-10-02

**Authors:** Caio Bezerra Machado, Flávia Melo Cunha de Pinho Pessoa, Emerson Lucena da Silva, Laudreísa da Costa Pantoja, Rodrigo Monteiro Ribeiro, Manoel Odorico de Moraes Filho, Maria Elisabete Amaral de Moraes, Raquel Carvalho Montenegro, Rommel Mário Rodriguez Burbano, André Salim Khayat, Caroline Aquino Moreira-Nunes

**Affiliations:** 1Pharmacogenetics Laboratory, Drug Research and Development Center (NPDM), Department of Medicine, Federal University of Ceará, Fortaleza 60430-275, Brazil; caio.bmachado97@gmail.com (C.B.M.); flaviamelop@outlook.com (F.M.C.d.P.P.); lucenaemerson@alu.ufc.br (E.L.d.S.); odorico@ufc.br (M.O.d.M.F.); betemora@ufc.br (M.E.A.d.M.); rmontenegro@ufc.br (R.C.M.); 2Department of Pediatrics, Octávio Lobo Children’s Hospital, Belém 60430-275, Brazil; laudreisa@hotmail.com; 3Oncology Research Center, Department of Biological Sciences, Federal University of Pará, Belém 66073-005, Brazil; rommel@ufpa.br (R.M.R.B.); andresk@ufpa.br (A.S.K.); 4Department of Hematology, Fortaleza General Hospital (HGF), Fortaleza 60150-160, Brazil; rmonteiroribeiro@icloud.com

**Keywords:** hematologic neoplasms, targeted molecular therapy, protein kinase inhibitors, TKIs

## Abstract

Cancer is still a major barrier to life expectancy increase worldwide, and hematologic neoplasms represent a relevant percentage of cancer incidence rates. Tumor dependence of continuous proliferative signals mediated through protein kinases overexpression instigated increased strategies of kinase inhibition in the oncologic practice over the last couple decades, and in this review, we focused our discussion on relevant clinical trials of the past five years that investigated kinase inhibitor (KI) usage in patients afflicted with relapsed/refractory (R/R) hematologic malignancies as well as in the pharmacological characteristics of available KIs and the dissertation about traditional chemotherapy treatment approaches and its hindrances. A trend towards investigations on KI usage for the treatment of chronic lymphoid leukemia and acute myeloid leukemia in R/R settings was observed, and it likely reflects the existence of already established treatment protocols for chronic myeloid leukemia and acute lymphoid leukemia patient cohorts. Overall, regimens of KI treatment are clinically manageable, and results are especially effective when allied with tumor genetic profiles, giving rise to encouraging future prospects of an era where chemotherapy-free treatment regimens are a reality for many oncologic patients.

## 1. Introduction

Cancer is a subset of noncommunicable diseases responsible for millions of deaths every year and is still seen as a major barrier to life expectancy increase worldwide [1,2]. Human tumors develop from healthy cellular populations after genetic deregulation at chromosomal or molecular levels that lead to increased cellular proliferation and overexpression of survival mechanisms. Complex interactions between heterogenic tumor cellular populations and tumor/human organism conceive biological advantages that ensure continuous growth in neoplastic clones, even in otherwise adverse scenarios [3,4].

Leukemias and lymphomas are a group of several hematologic and lymphoid tissue disorders that are characterized by accelerated expansion of clonal neoplastic populations of immunohematologic cell lines in the peripheral blood/bone marrow and lymph nodes of afflicted patients [5,6]. Non-Hodgkin’s lymphomas are among the 10 more incident cancers in the world, and, added together, leukemias afflicted more than 400,000 people every year, with diverse treatment efficacies for the different acute and chronic subtypes [2,6].

The ability of cancer cells to sustain continuous proliferative signals is a hallmark of cancer, and such ability is often dependent on the increased activities of growth factors and protein kinases (PK) [4]. Over the last couple decades, the advent of kinase inhibitor (KI) treatment in the oncologic practice represented a major step in targeted therapies development because it greatly enhanced the prognosis of many patients who used to depend on highly cytotoxic drug protocols for a chance of cancer treatment success [7,8].

In this study, we discuss the initial use of conventional chemotherapy treatment regimens and the emergent tumor resistance mechanisms, the development of KIs and their pharmacological characteristics as well as relevant clinical trials of the past five years that investigated KI usage in relapsed/refractory (R/R) hematologic malignancies.

### Conventional Therapies and Resistance in Hematologic Neoplasms

The process of self-renewal of hematopoietic stem cells is essential to homeostasis maintenance. Once this process is disrupted and defects in cell differentiation occur, the malignant transformation and the appearance of hematologic neoplasms can be triggered [9]. Although these neoplasms may have unclear etiology, genetic alterations as chromosomal rearrangements, point mutations, aneuploidies, deletions, insertions, duplications, and/or amplifications are characteristics of hematologic neoplasms [10,11,12,13].

Leukemias were first described in 1827, and since then, treatment has changed dramatically. First, studies showed that some cases were treated on single-agent chemotherapy by using nitrogen mustard in Hodgkin’s disease and other lymphomas and chronic leukemias, as well as by administration of the folic acid antagonist aminopterin in acute leukemia patients [14,15,16,17]. The first leukemia treatment protocol was a combination of radiation, arsenic and thorium-X [15,16] and, even after some decades, arsenic continued as part of leukemia treatment. A study published later demonstrated the successful effect of arsenic in the treatment of promyelocytic leukemia, once 11 patients presented clinical remission after arsenic administration [18].

Then, research and development of new classes of antitumoral agents such as antimetabolites, antibiotics, and alkaloids with promising results in pre-clinical and clinical tests were inserted in leukemia treatment protocols, first as single agents and after in combination, with the perspective of prolonging or maintaining disease remission [19]. For example, the 1950s and 1960s introduced several new drugs to antileukemic chemotherapy as the use of cortisone, chlorambucil, mercaptopurine, cytoxan, vincristine, vinblastin, daunorubicin, and bleomycin to the treatment of leukemias and/or lymphomas [20,21,22,23,24,25,26], methotrexate and vincristine in childhood acute leukemia [27,28], L-asparaginase and daunorubicin in acute leukemia therapy [29,30]. In 1958, methotrexate was administered intrathecally to kill leukemic cells present in the central nervous system [31] and chlorambucil and cyclophosphamide were later used in the treatment of chronic granulocytic and lymphocytic leukemias, respectively [32,33].

The combined protocols were introduced at the end of the 1960s, with the administration of cyclophosphamide, mercaptopurine, methotrexate, and vincristine in acute lymphocytic leukemia in children [34], cyclophosphamide and vinca alkaloids in malignant lymphoma [35] and cytarabine (AraC) and daunorubicin in acute myeloid leukemia [36]. With the increase in the number of antileukemic drugs and treatment protocols available, and with the better understanding of hematologic neoplasm development, the survival rate improved expressively among pediatric leukemia patients, as well as in patients with Hodgkin’s disease and the non-Hodgkin’s lymphomas [15,37,38,39].

However, over time, cases of drug resistance to available chemotherapy were diagnosed among patients with relapsed leukemias. Then, protocols with more intensive chemotherapy and radiotherapy were evaluated, with extensively toxic effects on patients [19,40,41]. In the 1970s, Don Thomas presents bone marrow transplants as a complement to acute leukemia treatment, with promising results, but with several obstacles related to tissue typing, infection control, immunosuppression, transfusion support and the need to develop more specific drugs that escape from cellular resistance [15,42,43].

There are several factors related to cancer cell resistance to antitumoral compounds described in the literature. Those can be divided into intrinsic factors that are pre-existent to the treatment, and extrinsic ones that are acquired after starting chemotherapy (Figure 1). Together, those factors are responsible for treatment failure, cancer recurrences in the clinic and enhanced mortality rates among affected patients [44,45].

Intrinsic resistance is characterized by resistance mechanisms inherent to the patient, once that is not triggered by the administration of chemotherapy drugs [46]. Factors such as (1) the existence of genetic mutations (mutations, gene amplifications, deletions, chromosomal rearrangements, transposition of the genetic elements, translocations, and alterations in microRNA expression), (2) presence of nonsensitive subpopulation (e.g., cancer stem cells) and (3) activation of mechanisms against xenobiotics (such as anticancer drugs) are responsible for reduced efficacy and treatment failure [46,47,48].

The acquired or extrinsic resistance occurs when the malignant cell line becomes less responsive to chemotherapy over time, that is, the antitumoral drug loses efficacy. The factors that lead to acquired resistance are grouped into the (4) activation of new proto-oncogenes, (5) mutations and/or alterations in transcription of drug targets, (6) modifications in the tumor microenvironment (TME) after the beginning of treatment [49,50,51].

In leukemias and lymphomas, some resistance mechanisms are more frequent than others. A study showed that the presence of cancer stem cells subpopulations in leukemia is related to less responsiveness to cytotoxicity chemotherapy, once these cells demonstrated a reduced proliferative rate [52]. In acute myeloid leukemia (AML), the treatment with the AraC is passive of acquired resistance. AraC is a pro-drug that needs to be phosphorylated to AraC-triphosphate to reach its drug target and reduced expression in metabolic pathways and mutations in AML cell lines leads to decreased levels of active drug, causing drug-resistance to AraC treatment [53,54].

Increased expression of the mechanism of drug efflux is well known as a cause of leukemia recurrence. Transmembrane transporters from the ATP Binding Cassette (ABC) superfamily ABCB1 (*MDR1*), ABCC1, ABCG2 are involved in the acquired resistance in AML, acute lymphocytic leukemia (ALL) and chronic myeloid leukemia (CML) [55,56,57].

CML is characterized by the presence of the Philadelphia chromosome (Ph+). The translocation between chromosomes 9 and 22 creates a chimeric protein BCR activator of RhoGEF and GTPase-ABL proto-oncogene 1 (BCR-ABL1), which has a tyrosine kinase activity [58,59]. The targeted therapy with tyrosine kinase inhibitors (TKIs) shows good therapeutical effect in CML patients, but mutations in CML patients genotypes are now the new challenge to TKIs therapy and in the CML clinic, once approximately 20–30% of patients do not respond to TKI from first-generation [58,60,61].

## 2. The Advent of Kinase Inhibition

Protein kinases (PK) are signaling regulators involved in various cellular functions including metabolism, cell cycle regulation, survival, and differentiation. Once activated, PKs typically phosphorylate serine, threonine or tyrosine residues on the target protein, leading to conformational change and consequent functional activation of the target proteins [62].

Phosphorylation of the target proteins by kinases is tightly regulated, and any perturbation to this regulation may lead to a diseased state. Multiple mechanisms lead to deregulation of kinases, enhancing oncogenic potential, which may include overexpression, relocation, fusions, point mutations and deregulation of upstream signaling (Figure 2). These discoveries led to the development of several KIs with wide applications in the clinical practice [62,63].

Imatinib was the first TKI to be approved by the US Food and Drug Administration (FDA) for the treatment of patients with CML. Imatinib inhibits the kinase activity of the fusion protein BCR-ABL1—encoded by a gene mutated in all patients with Ph+ CML—through a competitive mechanism at the ATP-binding site [64]. This blockage prevents the transduction of intracellular signals necessary for cell proliferation and apoptosis evasion. In addition, imatinib inhibits the proliferation of cells from different CML lineages and hematopoietic progenitor cells [65].

Imatinib (Glivec^®^) marked the beginning of the era of kinase inhibition as a reality for oncologic patients, although later on, the emergence of tumor resistance was unavoidable. After the introduction of imatinib in the early 2000s, the more selective second-generation drugs dasatinib, nilotinib, and bosutinib, followed by the third-generation compound ponatinib, enriched the therapeutic options to treat patients with Ph+ leukemias [66,67].

Approximately 20 to 30% of patients are bound to express resistance to imatinib. The mechanisms of resistance can be explained by mutations in the kinase domain of the BCR-ABL chimeric protein, gene amplification and overexpression of the BCR-ABL gene, alteration in the expression of influx and efflux transmembrane proteins and alterations in the regulation of signal transduction mechanisms. The T315I mutation in the BCR-ABL kinase domain is the most clinically alarming because only the third-generation inhibitor, ponatinib, has demonstrated efficiency in treating T315I-mutated tumors [68].

Dasatinib is a second-generation TKI approved for the treatment of CML and Ph+ ALL following imatinib treatment failure. Its range of targets include BCR-ABL, KIT proto-oncogene (c-Kit), platelet-derived growth factor receptor (PDGFR), and members of the SRC proto-oncogene (SRC) family [62,63,64]. Due to its ability to inhibit the proliferation of most mutant cells resistant to imatinib, dasatinib has been shown to be a good alternative for the treatment of patients who do not respond well to imatinib. However, a high toxicity of this drug was observed in patients in the more advanced stages of the disease due to the necessary high therapeutic doses [65].

Nilotinib is also a second-generation BCR-ABL kinase inhibitor. It has a similar spectrum of kinase targets that includes BCR-ABL, c-Kit and PDGFR and has activity against most imatinib resistance-conferring mutations, except the T315I mutation domain of the *BCR-ABL* gene, for which it remained ineffective. It was designed to overcome the imatinib resistance by binding to the kinase domain of imatinib-resistant mutants of BCR-ABL and imatinib-sensitive BCR-ABL with higher affinity [62,64,68].

The molecules that act as inhibitors of PKs catalytic activities are classified according to the mechanism of interaction with the kinase domain of target protein, being categorized into reversible and irreversible inhibitors. The reversible inhibitors are subclassified into types I, I½, II, III, IV or V, according to their connection and interaction with the PK domains [69].

Type I inhibitors are also known as “competitive ATP” inhibitors because they interact at the ATP binding site in the PK domain in its active conformation. The molecular recognition site of type I, I½ and II inhibitors is the hinge region. Examples of FDA-approved type I protein kinase inhibitors are gefitinib (epidermal growth factor receptor (EGFR) inhibitor), sunitinib (vascular endothelial growth factor receptor (VEGFR) and PDGFR inhibitor) and dasatinib (BCR-ABL inhibitor) [69].

Type I½ inhibitors interact with the activation segment of the kinase domain in a conformation that points towards the ATP-binding site (DFG-in) while type II inhibitors target the PK activation segment in the inactive conformation that points away from the ATP-binding site (DFG-out). Examples of FDA-approved type I½ and type II PK inhibitors are lapatinib (EGFR inhibitor) and nilotinib (inhibitor of BCR-ABL), respectively [69].

Type III and IV inhibitors are both allosteric in nature. While type III molecular recognition occurs exclusively at an allosteric site adjacent to the hinge, the type IV category refers to those allosteric inhibitors whose molecular recognition occur at a site distant from the hinge, neither exerting direct action over the ATP binding site. These inhibitors have the advantage of ensuring greater selectivity for the targeted proteins. Trametinib and cobimetinib are currently type III PK inhibitors approved by the FDA, both targeting mitogen-activated protein kinase kinase (MEK), and no type IV inhibitors have FDA approval to date. Type V inhibitors have bivalent activity, binding to two different regions of the PK domain [70].

The irreversible inhibitors are able to interact with the target protein through covalent bond formation and most recently have been categorized as type VI inhibitors. Examples of FDA-approved covalent and irreversible PK inhibitors are afatinib (erb-b2 receptor tyrosine kinase 2 (HER2) and EGFR inhibitor), ibrutinib (Bruton’s tyrosine kinase (BTK) inhibitor) and osimertinib (selective mutant T790M EGFR inhibitor) [70].

Kinase inhibition has revolutionized the practice of oncology and hematology for the past 20 years with over 40 compounds approved by the FDA. The therapeutic potential of kinase inhibition in oncology has been rapidly expanding beyond its origins in receptor tyrosine kinase oncogenes. The emergence of resistance mechanisms to existing kinase-targeting drugs has motivated a search for alternative targets, and the expansion of kinase inhibitor drugs into new target space continues to be facilitated by innovative strategies in precision medicine and molecular techniques [71,72].

## 3. Recent Prospects into Clinical Investigations

The usage of KIs in the treatment of several leukemia subtypes is an already established course of action in clinical practice. CML, FMS-like tyrosine kinase 3 (FLT3)-mutated AML and B-cell neoplasms are amongst the malignancies most responsive to KI treatment, including FDA approved drugs [73,74,75].

Although many inhibitors are used as first-line treatments, kinase inhibition may also be relevant as a secondary option after conventional chemotherapy resistance or relapse [76]. Table 1 is comprised of a series of clinical trials that, in the past five years, utilized KIs as a monotherapy or in combination with other cytotoxic agents to treat patients afflicted R/R hematologic disorders and results with varying degrees of efficacy were reported.

Of 32 clinical trials described in Table 1, 16 focused on patients afflicted with AML or other myeloproliferative disorders and had FLT3 as the main target of inhibition, with treatment regimens relying on first-generation—sorafenib—and second-generation—gilteritinib and quizartinib—inhibitors, and one study utilized the multikinase inhibitor pexidartinib. The other half of the analyzed studies focused primarily on lymphoid malignancies, especially of B-cell origin, and had phosphatidylinositol 3-kinase (PI3K) and Bruton’s tyrosine kinase (BTK) being the most targeted kinases (Figure 3). A large spectrum of different KIs and treatment protocols were covered, and kinase inhibition was mainly evaluated as a single-agent strategy in the treatment of R/R hematologic disorders, with only 10 out of 32 clinical trials associating KI usage with another cytotoxic agent [77,78,79,80,81,82,83,84,85,86,87,88,89,90,91,92,93,94,95,96,97,98,99,100,101,102,103,104,105,106,107,108].

It is important to note that, in the analyzed studies, no clinical trial investigating KI efficiency in CML or ALL cohorts was identified. This observation is likely due to the existence of already consolidated therapy options for both malignant subtypes. While CML patients highly benefit from the usage of imatinib and other second-generation TKIs, pediatric ALL patients achieve high probabilities of survival when treated under protocols of induction and consolidation therapies utilizing cytotoxic agents, with adult ALL patients adopting pediatric-inspired treatment regimens [109,110,111].

## 4. AML and Myeloproliferative Disorders

FLT3 is a receptor tyrosine kinase with a major role in hematopoiesis, being expressed in undifferentiated myeloid and lymphoid progenitors. Mutations in different FLT3 domains are associated with poor prognosis in AML patients and can be expressed as either internal tandem duplication (FLT3-ITD) or tyrosine kinase domain (FLT3-TKD) mutations. Usage of FLT3 inhibitors, as a monotherapy or in combination regimens, have demonstrated superior outcomes to standard chemo-immunotherapy and are a promising prospect for the future of AML treatments, even though impairments regarding tumor-acquired resistance and duration of response to the treatment are still relevant in the clinical practice [112,113,114].

Sorafenib is an FDA approved KI for the treatment of renal cell carcinomas and hepatocellular carcinomas that possesses multikinase inhibition proprieties, with VEGFR inhibition being the main focus of interest in clinical practice [115,116]. As a type II inhibitor, sorafenib binds to the activation loop of inactive VEGFR forms in a reversible manner by interacting with hydrophobic allosteric pockets deep within the kinase structure [117].

Although not a standard in clinical practice, sorafenib has a demonstrated activity in FLT3-ITD inhibition and has been used, with modest results achieved, as a monotherapy for the treatment of AML patients in previous clinical trials [118,119]. While, in the analyzed studies, usage of sorafenib in combination therapies for the treatment of R/R myeloproliferative disorders yielded encouraging results based on response rates, parameters of patient overall survival and progression-free survival are still generally poor and may relate to inhibitor’s inability to induce deep molecular response and reduction of FLT3-ITD allelic burden in all treated patients [80,87].

Clinical trials from the past year also reassured sorafenib’s importance in AML treatment. Burchert et al. and Xuan et al. demonstrated, through individual studies, that usage of sorafenib by patients afflicted with FLT3-ITD AML after allogeneic hematopoietic stem-cell transplantation increases relapse-free survival while presenting minimal toxicities when compared to the placebo group [120,121].

Gilteritinib and quizartinib are both second-generation inhibitors, the former being a type I inhibitor and the latter being type II, with high selectivity for FLT3. While only gilteritinib has FDA approval for the treatment of R/R AML, studies utilizing either inhibitor for the treatment of AML FLT3-mutated patients have already described increased response rates over traditional salvage chemotherapy [122,123,124].

The problem around quizartinib approval for clinical use in AML management is the considerably low benefit-ratio due to the emergence of tumor TKD-mutation-mediated resistance. FLT3-TKD mutations frequently occur at the kinase activation loop and lead to constitutively active kinases which are not suitable targets for type II inhibitors because this subclass is dependent on the inactive protein conformation to be able to adequately bind around the ATP sites. As a type I inhibitor, which binds to active kinase conformations, gilteritinib is able to avoid resistance mechanisms that hinder type II inhibitor activity, and it demonstrates increased efficiency as a monotherapy in R/R AML patients [124,125,126].

Two clinical trials evaluating gilteritinib as a monotherapy reported increased treatment efficacy, which translates to higher overall response rates (ORR) and complete remissions when applied to patients whose FLT3 mutation-positive status was known compared to patients with wild type FLT3 [100,104]. Such results may be attributed to the increased specificity of gilteritinib, as a second-generation inhibitor, in targeting FLT3 when compared to KIs such as sorafenib, which, in fact, have multikinase activities and may also regulate tumorigenesis through other distinct molecular pathways [127].

Patients suffering from myeloproliferative disorders under treatment regimens that included KIs with molecular targets other than FLT3 inhibition did not report significant clinical responses and parameters of survival and disease progression were generally poor. However, the occurrence of adverse events (AE) was not drastically different from what is reported of most KI therapies, indicating a tolerable profile that may permit further studies in the investigation of their association with synergetic compounds [92,95,98,99,105,107,108].

An exception to the above-mentioned inefficacy of KIs other than FLT3 inhibitors was the activity of rigosertib, an inhibitor of RAS signaling pathways, when used alone or in combination with azacitidine, a nucleoside analog, for the treatment of myelodysplastic syndrome (MDS) patients. The ORR and bone marrow responses for this patient cohort demonstrated encouraging results, and the observed AEs did not represent an impairment for continuous therapy, even when analyzed in a drug combination regimen [81,103].

## 5. Lymphoid Malignancies

PI3K is a family of protein kinases that act as second messengers downstream of receptor tyrosine kinases and G-protein-coupled receptors. Their activity in cellular proliferation and metabolism is highly associated with cascade activation of AKT serine/threonine kinase (AKT)/mechanistic target of rapamycin kinase (mTOR) pathways, although PI3K AKT/mTOR-independent mechanisms are also relevant to cancer development [128,129].

While its well-defined role in carcinogenesis puts a spotlight into PI3K inhibition in the oncological practice, the many physiological cellular routes associated with PI3K/AKT/mTOR pathway represent an impairment to its proper targeting in cancer due to an elevated number of related toxicities and off-target effects [130]. A priority in the development of selective PI3K inhibitors with activity on specific kinase isoforms has been seen in the recent years over the development of pan-PI3K inhibitors, which are responsible for a broader spectrum of treatment related AEs [131].

Idelalisib is a selective PI3K-δ isoform inhibitor and was the first PI3K inhibitor to receive FDA approval, being indicated to treat R/R chronic lymphocytic leukemia/small lymphocytic lymphoma (CLL/SLL) and R/R follicular lymphoma (FL) [132]. PI3K-δ is highly expressed in malignant B-cells, being associated with tumor proliferation and apoptosis evasion, and idelalisib’s inhibition over PI3K-δ regulates the downstream activities of B-cell receptor pathways, which are also main effectors in B-cell malignancies pathogenesis [133]. Other relevant PI3K inhibitors with FDA approval to treat hematological malignancies include the pan-PI3K inhibitor with preferential activity towards PI3K-α/-δ copanlisib, the PI3K-γ/-δ inhibitor duvelisib and the recently approved PI3K-δ/Casein kinase 1 epsilon (CSNK1E) inhibitor umbralisib [134,135,136].

The high prevalence of clinical trials evaluating PI3K inhibition as therapeutics for B-cell malignancies speaks to the favorable outcomes, especially when combined with chemo-immunotherapy treatment regimens, achieved in these studies, with ORRs reaching results as high as 75% of the treated population. Treatment efficacy, however, is diverse among different malignant B-cell subtypes, and patients afflicted with R/R diffuse large B-cell lymphoma (DLBCL) had generally lower rates of response to PI3K inhibition. Even among DLBCL patients, molecular profiles distinguishing the cell of origin in activated B-cell-like (ABC) DLBCL and germinal center B-cell-like (GCB) DLBCL represent a further stratification when predicting patient outcome to PI3K inhibition treatment [85,88,90,94,96,97,102,106].

Mechanisms involved in tumor-acquired PI3K-inhibitor resistance are not fully elucidated yet, with no common mutation characterized across patient cohorts with progressive disease after idelalisib treatment [137]. However, analyses in human and murine models signal towards upregulation of mitogen-activated protein kinase (MAPK)/extracellular signal-regulated kinases (ERK) pathways in neoplastic cells resistant to PI3K-δ inhibition, which are major cellular mechanisms responsible for the regulation of proliferation, differentiation and cell death [138,139,140]. In the murine model, MAPK/ERK activity was enhanced due to overexpression of insulin-like growth factor 1 receptor (IGF1R) and concomitant treatment with linsitinib, an IGF1R inhibitor, was able to overcome PI3K-δ resistance, indicating a possibility for investigation into combination treatments in the clinical practice [140].

Another highly relevant kinase in the development of B-cell neoplasms is BTK, and as such, it was also a prevalent molecular target in the aforementioned studies. BTK is a nonreceptor tyrosine kinase that is a member of the Tec protein tyrosine kinase (TEC) family and, alongside PI3K, BTK is a main effector of downstream B-cell receptor signaling pathways, playing a critical role in proliferation and metabolism of B-cells as well as in their carcinogenesis [141,142].

Ibrutinib is an oral and irreversible first-generation BTK inhibitor that revolutionized CLL treatment since its original FDA approval in 2013 [143]. Ibrutinib binds covalently to a cysteine residue (Cys481) in the BTK active domain, hindering kinase activity and also regulating downstream pathways. The bond formed to Cys481, however, is not completely selective to BTK and may promote ibrutinib’s activity over other off-target kinases, such as other members of the TEC family or EGFR, increasing the occurrence of treatment-related AEs and toxicities [144,145].

Although a standard of care in many lymphoid malignancies, primary and acquired resistances to ibrutinib treatment are still problematic in the oncologic routine. The specific C481S mutation, associated with a cysteine-to-serine exchange in BTK active domain, is well characterized as altering ibrutinib’s capability to covalently bind to BTK and handicapping its effectiveness over tumor-cell proliferation. Overexpression of distinct cell survival mechanisms, tumor microenvironment and cancer stem cell metabolism have also been indicated as possible routes for tumor resistance to ibrutinib in lymphoid neoplasms [146,147,148].

Acalabrutinib, zanubrutinib and tirabrutinib are second-generation irreversible inhibitors that present much more selective activity towards BTK than towards other TEC family kinases, with FDA approval to treat B-cell malignancies encompassing only the first two [149,150]. While clinical investigations into second-generation BTK inhibitors are prevalent, their mechanisms of action do not seem to overcome ibrutinib treatment resistance, but still present advantages when analyzing treatment-related AEs. Clinical trials comparing acalabrutinib or zanubrutinib efficacy over ibrutinib in patients afflicted with R/R lymphoid malignancies determined that, while their response rates are similar, the usage of second-generation inhibitors relate to much more tolerable toxicity profiles with special emphasis on cardiovascular tolerability [150,151,152].

First- and second-generation BTK inhibitors, as a monotherapy or in combination protocols, have been evaluated in clinical trials for the past half-decade, and the efficacy results verify them as a solid choice for the treatment of B-cell malignancies even as a second-line strategy, achieving exceedingly high response rates in most studies and allowing for long periods of progression-free survival and overall survival. These observed results are in line with what is expected from the utilization of BTK inhibitors because, although relatively recent in clinical use, their major role in lymphoid malignancies management is already a standard [79,82,84,86,89,93,96].

Both PI3K and BTK are well established molecular targets in cancer and count with FDA approved drugs for the treatment of an array of malignant subtypes, but still deal with emerging cases of drug resistance and inability to promote complete remission when used as monotherapy for R/R malignancies. In this context, a study conducted by Davids et al. investigated the outcome of patients under ibrutinib, a first-generation BTK inhibitor, and umbrasilib, a selective PI3K-δ inhibitor, combination therapy and determined that this treatment protocol is not only effective but also clinically safe and warrants further investigation to fully elucidate its potential in the clinical practice [96,153,154].

## 6. Toxicity Profiles

Irrespective of associated treatment, correlations in the appearance of AEs may be observed. Most studies described in Table 1 reported grade ≥3 hematologic AEs that may be summarized as neutropenia, anemia and thrombocytopenia, and these findings are not unexpected because they are in accordance with the described toxicity of many KIs [155]. Other nonhematologic AEs observed in these studies are mainly associated with gastrointestinal disorders, primarily diarrhea and nausea, and are also commonly reported toxicities of KI usage that may be related to the important role of kinases in protein phosphorylation during mitosis and normal cell metabolism [155,156].

Specific AEs related to certain KI subclasses are also reported. In both studies analyzing sorafenib treatment, association with hand-foot skin reactions (HFSR) and rashes represent dose-limiting toxicities and are managed by dose reductions or treatment interruption [80,87]. HFSR represents a major problematic when using sorafenib to treat patients afflicted with advanced hepatocellular carcinoma. The mechanisms through which HFSR manifests are not yet fully elucidated, and management is still complicated as dose reductions are not always possible because it could promote disease progression [157]. Future clinical trials evaluating sorafenib as an option for hematologic malignancies should have in mind that skin reactions are a common treatment-related AE and further investigations into management options are still warranted.

Second-generation FLT3 inhibitors, quizartinib and gilteritinib, are more selective and toxicity profiles observed are more tolerable, with less off-target side effects. However, quizartinib treatment is associated with QT interval prolongation that is mostly manageable without dose reductions, while gilteritinib is related to elevated alanine aminotransferase and aspartate aminotransferase levels and increased blood creatine phosphokinase and lactate dehydrogenase [91,100,101,104].

As a drug class, PI3K inhibitors exhibit an array of associated toxicities due to side effects related with PI3K/AKT/mTOR signaling in many cellular pathways [158,159]. Copanlisib, an inhibitor of pan-PI3K activity, presents elevated rates of grade ≥3 AEs and serious AEs, and a relevant emphasis must be given on hypertension and hyperglycemia rates among treated populations that reflects copanlisib’s on-target activity over PI3K-γ and PI3K-α isoforms, respectively, but emergent AEs are generally predictable and may be easily managed [85,130,159].

PI3K inhibitors with selective activity towards isoform δ tend to exhibit more immune-related toxicities, due to isoform’s specific expression in lymphocytes [160]. Besides hematologic toxicities, idelalisib and duvelisib treatment are associated with infections and immune response disorders such as pneumonia, urinary tract infections, upper respiratory tract infections and colitis [90,97,106]. Of the previously described studies, umbralisib was the PI3K-δ selective inhibitor with less severe associated toxicities and grade ≥3 AEs mostly manifested as neutropenia [88,102].

Patients treated with BTK inhibitors are prone to diverse AEs of which some molecular mechanisms are not completely understood. Cutaneous side-effects, minor and major bleedings, cardiac toxicities and increased susceptibility to infections are among the previously described treatment-related AEs [161,162]. Of the previously described studies, ibrutinib treatment was indeed a risk factor for atrial fibrillation occurrence and hypertension as well as major hemorrhages and infections such as pneumonia, urinary tract infections, cellulitis, sepsis and lung infections. AE prevalence tends to trend down for patients on continuous therapy over the years, with the exception of hypertension and bruising, both of which trended upwards [89].

The usage of second-generation inhibitors acalabrutinib and zanubrutinib is more selective in targeting BTK and is associated with lesser cardiovascular toxicities, which are hypothesized to be related to ibrutinib’s inhibition of PI3K and B-cell receptor cascade signaling pathways [162,163]. However, the toxicity profiles characterizing increased infection susceptibility tend to remain the same among different BTK inhibitors and high incidence of pneumonia and upper respiratory tract infections are still observed even in second-generation treatment settings [79,82,84,86].

Targeted therapies focused towards B-cell receptor signaling disruption, such as ibrutinib or idelalisib, have been associated with lymphocytosis in early stages of B-cell malignancies treatment, and, while it may be seen in the treated population as a whole, increases in lymphocyte count in peripheral blood are more prominent in R/R malignant settings and in immunoglobulin heavy variable cluster (IGHV)-mutated patients. It is imperative that professionals in the oncologic practice understand that this manifestation is not considered to be an AE or a bad prognostic factor because these lymphocyte populations have been demonstrated as not actively proliferative and thus not being neoplastic clones, more likely representing lymphocytes migrating from lymphoid tissues to the blood stream [164,165,166].

Even though toxicity profiles are still relevant when evaluating KIs efficiency, the development of second-generation inhibitors with more specific targeted activity and consequent less off-targeted toxicities—as well as the demonstrated increased response rates over traditional chemo-immunotherapy when treating kinase-mutated tumors—fuels the rationale for KIs accelerated development and increased clinical use, especially when associated with a tumor’s genetic profile [167,168].

## 7. Conclusions

The utilization of kinase inhibitors as an alternative to traditional chemo-immunotherapy in the oncological practice still face obstacles in the nonspecific activity of many first-generation inhibitors, the need to be associated with a genetic profile in order to achieve ideal outcomes and the emergence of tumor resistances that may impair inhibitors’ molecular pathways. However, in the last decades, investigations into their efficacy and the positive outcomes achieved in the clinical practice give rise to encouraging future prospects of an era where chemotherapy-free treatment regimens are a reality for many oncologic patients as new options for first- and second-line targeted therapies continue to emerge.

## Figures and Tables

**Figure 1 pharmaceutics-13-01604-f001:**
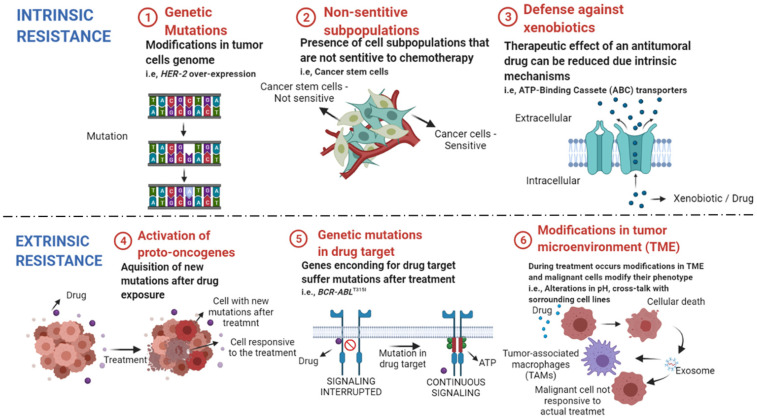
Intrinsic and extrinsic mechanisms of drug resistance in cancer. ① Genetic mutations can lead to alterations in the expression of genes related to cellular resistance and surveillance. ② Heterogenous tumors have subpopulations that may not respond to the available cytotoxic drugs, leading to cancer remission after treatment. ③ Some cellular transporters protect our cells from environmental toxins, as anticancer drugs, and reduce the concentration of their intracellular levels. ④ After treatment, new oncogenes can be activated, leading to enhanced proliferation rate of not responsive cells. ⑤ Mutations in genes that encode drug targets reduce drug efficacy in mutated cell lines. ⑥ Treatment can alter the tumor microenvironment (TME) and lead to cross-talk between sensitive cells and the surrounding cells. The exchange of resistance elements with tumor-associated macrophages (TAMs) and other tumoral cell lines lead to enhanced cell resistance to chemotherapy. Created with BioRender.com, accessed date: 20 August 2021.

**Figure 2 pharmaceutics-13-01604-f002:**
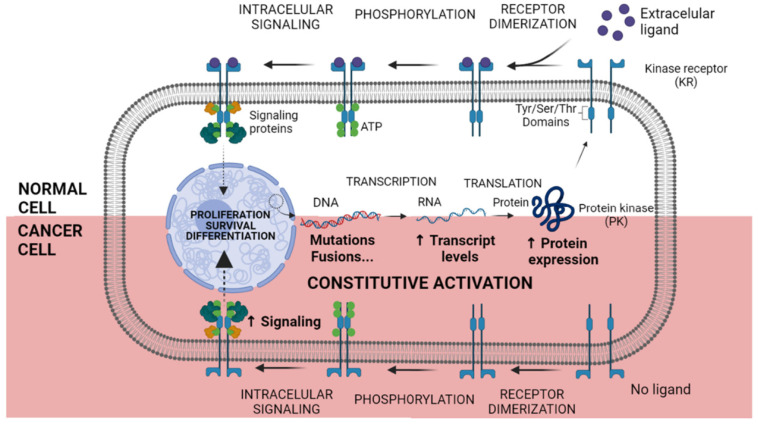
Constitutive activation of Kinase Receptor mechanism. In normal cells, after transcription and translation, Kinase Receptors (KR) are sent to the cellular membrane. In the presence of an extracellular ligand, the KR dimerization occurs and the Tyrosine (Tyr), Serine (Ser) or Threonine (Thr) domains are autophosphorylated. Signaling proteins are attached to the phosphorylated amino acid residues, and the downstream signalization begins. Otherwise, in cancer cells, genetic and chromosomal alterations lead to enhanced RNA and protein overexpression. Malignant cancer cells do not depend on an extracellular ligand to initiate the signaling process, being constitutively active and boosting tumoral phenotype. Created with BioRender.com, accessed date: 20 August 2021.

**Figure 3 pharmaceutics-13-01604-f003:**
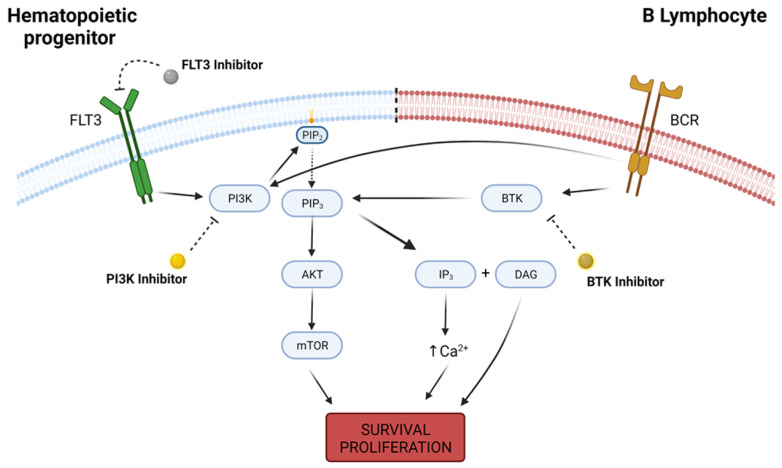
Intracellular pathways of FLT3, PI3K and BTK. FLT3 is a receptor tyrosine kinase highly expressed in hematopoietic progenitor cells, and its activity leads to downstream activation of survival pathways such as PI3K, which in turn converts phosphatidylinositol 4,5-bisphosphate (PIP2) into phosphatidylinositol-3,4,5-trisphosphate (PIP3) and recruits AKT to the cell membrane, with further upregulation in mTOR activity. PI3K/AKT/mTOR pathway is responsible for a wide variety of physiological functions that determine cellular survival and proliferation. On B lymphocites, B-cell receptor (BCR) is a transmembrane immunoglobulin that, upon activation, signals for PI3K and BTK activity. BTK regulates PIP3 degradation into IP3 (inositol 1-4-5 trisphosphate) and DAG (diacylglycerol), increasing intracellular calcium concentration and also promoting cellular survival and proliferation. Deregulation in the expression of any of these kinases may lead to malignant cell phenotype, and kinase inhibitors are a possible therapeutic option in the oncological practice. Created with BioRender.com, accessed date: 20 August 2021.

**Table 1 pharmaceutics-13-01604-t001:** Clinical trials utilizing kinase inhibitors (KI) as therapeutics for relapsed/refractory hematological malignancies in the past five years.

Clinical Study Phase	Leukemia Subtype	Targeted Kinase	Kinase Inhibitor	Associated Treatment	Clinical Outcome	Adverse Events	References
II	HCL	BRAF	Vemurafenib, 960 mg twice daily, orally administered for a total of 8 weeks	Eight intravenous rituximab doses (375 mg per square meter of body surface) over 18 weeks	Complete response achieved in 87% of the patients with PFS of 78% at a median follow-up of 37 months	Mostly grade 1 or 2 involving neutropenia, cutaneous rash, photosensitivity, fever, fatigue and others; Liver and pancreatic biochemical alterations were also detected	[77]
I	MM	PIM	PIM447, 250 mg or 300 mg, orally administered daily in a 28 continuous days cycle	NR	ORR was 15.4%, DCR was 69.2%, and CBR was 23.1%; All patients discontinued treatment due to physician decision or progressive disease	At least one grade 3 or 4 AE in all of the studied patients; The most common AEs associated with the treatment were thrombocytopenia, anemia, leukopenia and lymphopenia	[78]
Ib	CLL/SLL; FL	BTK	Zanubrutinib, 160 mg twice daily or 320 mg once daily in a 28 continuous days cycle	Obinutuzumab intravenously administered in 6 28-day cycles at different doses dependent on the cycle day	R/R CLL/SLL patients had a 92% ORR with 28% CR and median PFS was not reached; R/R FL patients had a 72% ORR with 39% CR and PFS of 25 months	The most common reported AE was upper respiratory tract infections in both patient cohorts; Neutropenia was the most common grade ≥3 AE; Only 6.17% of patients went through treatment discontinuation due to AEs	[79]
I	AML	FLT3	Sorafenib orally administered in 400 mg, 600 mg and 800 mg doses twice daily in a 28 continuous days cycle	G-CSF and plerixafor at 10 mcg/kg and 240 mcg/kg subcutaneous doses, respectively; Administration occurred every other day for seven doses starting on day 1	36% of patients responded to treatment with a median duration of response of 5.3 months; Most patients that responded remained FLT3-ITD positive during therapy	71.4% of patients presented grade ≥3 nonhematologic AE beyond cycle 1 including skin associated toxicities, cardiac arrhythmias, liver enzyme elevations, bone pain and others	[80]
I	MDS; CMML; AML	RAS	Rigosertib in 140 mg or 280 mg doses twice daily or in a total 840 mg daily dose in a 4-week cycle consisting in 3 weeks of treatment and 1 week of rest	Azacitidine in subcutaneous or intravenous doses of 75 mg/m^2^/kg for seven days starting one week after rigosertib initial treatment	Responses to treatment were achieved in 56% of patients with a median duration of response of 5.8 months;	89% of the patients experienced grade ≥3 AE including pneumonia, neutropenia and thrombocytopenia; 72% reported serious AEs, but none were considered related to study treatment; 33% reported genitourinary AEs of mostly grade 1 or 2	[81]
II	CLL/SLL	BTK	Zanubrutinib,160 mg twice daily in 28-day cycles	NR	84.6% of patients achieved a response, but only 3.3% achieved CR; 92.9% of responders did not have disease progression at 12 months follow-up	63.7% of patients reported grade 3 AEs, 8.8% reported grade 4 AEs and 3.3% reported grade 5 AEs. The most common AEs were neutropenia, upper respiratory tract infections, petechiae, anemia and hematuria	[82]
I/II	AML	FLT3	Pexidartinib orally administered twice daily in a dose expanding protocol of 800 mg to 5000 mg in a 28-day cycle; Pexidartinib in 3000 mg doses in phase II	NR	ORR for all treated patients was 21%; Patients in phase II had a median duration of response of 76 days, median PFS of 48 days and median OS of 112 days; Parameters were highly enhanced in responders	Most common grade ≥3 AEs were febrile neutropenia and anemia while the most common general AEs included diarrhea, fatigue and nausea; Only 1 patient experienced a fatal AE that was considered related to treatment; MTD was not reached	[83]
II	CLL/SLL	BTK	Acalabrutinib, 100 mg doses twice daily or 200 mg doses once daily each in 28-day cycles	NR	Median time for initial response was 5.5 months; ORR for all patients was 87.5%; Estimated PFS at 24 months was 84.3% for R/R patients	Most common AEs were grade 1 or 2 and included headache, contusion, diarrhea and upper respiratory tract infection; Grade ≥3 AEs occurred in 43.8% of patients and manifested primarily as neutropenia; One patient experienced grade 5 liver failure considered related to treatment	[84]
II	DLBCL	PI3K	Copanlisib in a 60 mg dose as an IV infusion in days 1, 8 and 15 of a 28-day cycle	NR	ORR was 19.4% with CR corresponding to 7.5% of patients; Median PFS in the overall cohort was 1.8 months and median OS was 7.4 months	97% of patients experienced some kind of AE; 86.6% of AEs were grade ≥3, the main ones being hypertension and hyperglycemia, and 65.7% of patients experienced serious AEs; Drug-related AEs leading to treatment discontinuation occurred in 11.9% of patients	[85]
I/II	CLL/SLL	BTK	Acalabrutinib in dose escalation of 100 mg to 400 mg once daily or 200 mg twice daily; Acalabrutinib in doses of 100 mg twice daily or 200 mg once daily in phase II	NR	ORR was 94% and CR was 4%; Median time to initial response of PR or better was 4.7 months; Estimated PFS at 45 months was 62%	AEs occurring in ≥10% of patients were generally mild to moderate and included diarrhea, headache, upper respiratory tract infection and fatigue; Grade ≥3 AEs occurred in 66% of patients and were mainly neutropenia and pneumonia; 10 patients (7.5%) had AE-related deaths	[86]
II	AML	FLT3	Sorafenib administered in 400 mg doses twice daily in a 21-day cycle	OME administered IV from day 1 to 7 in 2 mg/d doses; After CR/CRi, OME was administered from days 1 to 5 in new cycles	71.8% of R/R patients achieved CR/CRi after 1 or 2 cycles; Patients who achieved CR/CRi and did not undergo hematopoietic stem cell transplant relapsed despite continuous treatment; OS in nonresponders was shorter than 11 months while median OS in those who achieved CR/CRi was 10.9 months	Most AEs were grade 2 or lower and mainly included fever, rash and anemia; Grade ≥3 AEs included neutropenia, thrombocytopenia and hematuria	[87]
I/Ib	B-NHL; CLL/SLL	PI3K-δ	Umbralisib orally administered once daily at doses of 800 mg or 1200 mg (initial formulation) or escalating doses of 400 mg to 1200 mg (micronized formulation)	Ublituximab doses of 900 mg to B-NHL patients and 600 mg or 900 mg to CLL patients administered IV on different days according to cycle progression	ORR was 46% and CR was 17% with median time to first response of 8 weeks; Virtually all patients who responded (29 of 32) received therapeutic-dose levels of umbralisib	The majority of AEs were grade 1 or 2 and the most common were diarrhea, nausea and fatigue; Neutropenia was the most common grade ≥3 AE	[88]
III	CLL/SLL	BTK	Ibrutinib orally administered in doses of 420 mg daily	NR	ORR was 91% and CR/CRi was 11%; Median OS was 67.7 months and PFS was 44.1 months	Commonly reported grade ≥3 hematologic AEs were neutropenia, thrombocytopenia and anemia while nonhematologic AEs included pneumonia and hypertension; 10% of patients experienced major hemorrhage; Prevalence of AEs decreased over time for patients on continuous therapy	[89]
I	NHL; CLL	PI3K-δ; PI3K-γ	Duvelisib orally administered twice a day in 25 mg daily doses in a 28-day cycle	Rituximab (375 mg/m^2^ IV) on cycle day one (Arm 1); Rituximab (375 mg/m^2^ IV) with addition of variable doses of Bendamustine (Arm 2)	ORR in Arm 1 was 78.3% compared to 62.5% in Arm 2; Overall median PFS was 13.7 months and median OS of 9.1 months was only reached for NHL patients in Arm 2	95.7% of patients were afflicted by an AE considered related to treatment; 87% of patients experienced grade ≥3, the most common being neutropenia; Serious AEs related to treatment happened on 25.9% of patients in Arm 1 and 10.5% of patients in Arm 2	[90]
II	AML	FLT3	Quizartinib in daily doses of 20, 30 or 60 mg that increased by one dose level at a time in patients who did not achieve CRc	NR	CRc rate was 53.8%; Median duration of CRc was 16.1 weeks and median OS was 34.1 weeks	The most common AEs, as well as most common grade ≥3, were febrile neutropenia and platelet count decrease; Serious AEs were reported in 45.9% of patients	[91]
II	AML	VEGFR	Pazopanib in doses of 800 mg once daily	NR	PR was achieved in 10% of patients and was the best reported response; SD was reported in 70% of patients; Median PFS was 65 days and median OS was 191 days	Majority of AEs were gastrointestinal and included nausea, diarrhea and decreased appetite; The most common grade 3 AE was nausea and no grade ≥4 was reported; No serious AEs were considered related to treatment	[92]
I	B-NHL; CLL/SLL	BTK	Tirabrutinib administered in 160 mg, 320 mg or 480 mg once daily or 300 mg twice daily	NR	ORR for all cohorts was 76.5%, with variations when analyzing specific treatment cohorts and malignant subtypes; 12 out of 16 patients showed ≥50% reduction in tumor diameter	One DLT was observed in the 300 mg twice daily cohort; Most common AEs were rash, vomiting and neutropenia; Grade ≥3 AEs mainly included hematologic toxicities; 4 serious AEs related to treatment were reported	[93]
I/II	B-NHL	PI3K-δ; JAK 1	Parsaclisib in dose escalation that was later amended to doses of 20 mg once daily for the first 9 weeks, followed by parsaclisib 20 mg once weekly	Itacitinib 300 mg once daily or chemotherapy (rituximab, ifosfamide, carboplatin and etoposide)	As a monotherapy, ORR varied by disease type, with worse response of DLBCL patients (30%) and best response of MZL patients (78%); Durable responses were observed in patients following the once-weekly dosing schedule; In combination therapies, response rates varied highly and were inconsistent due to small amount of patients per group	As a monotherapy, the most common nonhematologic AE were diarrhea, nausea, fatigue and rash, while any-grade neutropenia was experienced by 44% of patients; In treatment with itacitinib, 45% of patients experienced grade 3/4 AEs and in chemotherapy combination, patients reported grade 4 thrombocytopenia and neutropenia	[94]
II	AML	MEK 1/2	Binimetinib administered in doses of 30 mg or 45 mg twice daily	NR	Only one patient (8%) achieved a CRi; Median OS was 1.8 months	The most common AEs were diarrhea, hypokalemia, hypotension, hypoalbuminemia and hypocalcemia; Only one serious AE related to treatment was identified; No treatment-related deaths occured	[95]
I/Ib	CLL; MCL	PI3K-δ; BTK	Umbralisib orally at doses of 400 mg, 600 mg or 800 mg; Ibrutinib at doses of 420 mg for CLL patients and 560 mg for MCL patients	NR	ORR for CLL patients was 90%; CLL 2-years PFS and OS were 90% and 95%, respectively; ORR for MCL patients was 67%; MCL 2-years PFS and OS were 49% and 58%, respectively; Median time to best response was 2 months in both cohorts	Common AEs included diarrhea, infection and nausea; Grade 3/4 hematologic AEs were not common and included neutropenia, thrombocytopenia and anemia, but were not considered related to study drugs; Serious AEs were experienced by 29% of patients	[96]
III	CLL/SLL	PI3K-δ; PI3K-γ	Duvelisib twice daily in doses of 25 mg in 28-day cycles	NR	Median PFS was 13.3 months and estimated 12-month PFS was 60%; ORR was 73.8% with PR being 72.5% of cases; Estimated 12-month OS was 86%	Most common hematologic AEs included neutropenia, anemia and thrombocytopenia; Grade ≥3 AE occurred in 87% of patients and 4 serious AEs experienced by patients were considered related to study drug	[97]
I	MDS; CMML	JAK 1/2	Ruxolitinib in doses of 5, 10, 15 or 20 mg twice a day in 28-day cycles	NR	Responses were achieved in 4 out of 18 patients, but 2 patients relapsed after 1 and 4 months; Median OS was 14.5 months with different outcomes depending on disease cohort	Most common nonhematologic AEs were hyperglycemia, fatigue and elevated liver function; Thrombocytopenia and anemia were the most common hematologic AEs and were also observed as grade ≥3	[98]
II	AML	SYK	Erlotinib in daily doses of 150 mg	NR	ORR was 10% with only one patient (3%) achieving CR; All patients discontinued therapy due to PD and median OS was 3.5 months	AEs considered related to study drug included fatigue, diarrhea, nausea and rash; Only 7% of patients had AEs that required dose reduction and treatment discontinuation	[99]
I	AML	FLT3; AXL	Gilteritinib in escalating doses of 20 to 300 mg daily	NR	ORR was 47.4% with CRc being 36.8% of overall cases; ORR and CRc rates were enhanced in patients who were FLT3 mutation-postive	DLTs were reported at 120 mg and 300 mg cohorts and MTD was established at 200 mg/day; Most drug-related AEs were elevated liver enzyme levels, elevated blood creatine phosphokinase and elevated blood lactate dehydrogenase; Serious AEs were experienced by 29.2% of patients	[100]
IIb	AML	FLT3	Quizartinib in daily doses of 30 mg or 60 mg in 28-day cycles	NR	Overall CRc was 47.4% and ORR was 61% and 71% for patients in the 30 mg and 60 mg groups, respectively; Duration of CRc and OS was longer on the 60 mg group	AEs considered related to treatment were reported evenly among both treatment groups and the most common included hematologic events, diarrhea and fatigue; Serious AEs considered related to treatment occurred in 26% of patients in 30 mg group and 22% of patients in 60 mg group	[101]
I	CLL/SLL; B-NHL; T-NHL; HL	PI3K-δ	Umbralisib daily in 28-day cycles following a dose escalation protocol to a maximum of 1800 mg a day; Later amendments transitioned all doses to 800 mg a day	NR	Of all patients treated, 62% reported reduction in disease burden, 33% had an objective response and 30% had a partial response; Efficacy varied highly among disease subtypes and best parameters of ORR, DOR and PFS were achieved in the CLL patients subgroup	DLTs were observed in cohorts of 800 mg and 1800 mg and the MTD was determined to be 1200 mg; Most common AEs were diarrhea, nausea and fatigue and were generally grade 1 or 2; Grade ≥3 AE included neutropenia, anemia and thrombocytopenia; Discontinuation of treatment due to AEs occurred in 7% of patients	[102]
I/II	MDS; AML	RAS	Rigosertib in continuous IV infusion at initial doses of 650 mg/m^2^ for 3 days on 14-day cycles; Dose escalation was possible depending on toxicity and effectiveness	NR	26.5% of patients achieved bone marrow/peripheral blood response and the same amount of patients achieved SD; Median OS for responding patients was 15.7 months in contrast to OS of 2 months for nonresponders	MTD was determined as 1700 mg/m^2^ and recommended phase 2 dosage was 1350 mg/m^2^; Most common AEs were fatigue, diarrhea and pyrexia; Most common grade ≥3 AEs were anemia, thrombocytopenia and pneumonia; 18% of patients had serious AEs related to treatment and 59% discontinued treatment due to AEs	[103]
I/II	AML	FLT3	Gilteritinib in dose escalation cohorts of 20 to 450 mg daily	NR	ORR was 40% and most CRc were achieved in patients in the 120 mg/day and 200 mg/day cohorts; Median OS for all patients was 25 weeks; Better ORR was observed for patients’ FLT3 mutation-positive	Grade ≥3 AE included febrile neutropenia, anemia, thrombocytopenia, sepsis and pneumonia; Drug-related AEs leading to treatment discontinuation happened in 10% of patients; MTD was defined as 300 mg/day	[104]
I	AML	AURK A/B	AMG 900 was administered, after protocol amendment, at escalating doses of 30 to 75 mg daily for 7 days every 2 weeks	NR	9% of patients, following the protocol before amendment, achieved CRi and no other responses were observed; For responders, maximum duration of response was 3 months	The most common AEs were nausea, diarrhea, febrile neutropenia and fatigue and the most common grade ≥3 AE was neutropenia; 31% of patients had serious AEs and 14% discontinued treatment due to AEs; Two deaths, respiratory failure and septic shock, were considered related to treatment	[105]
III	CLL	PI3K-δ	Idelalisib 150 mg twice daily	Ofatumumab administered IV in doses of 300 mg on day 1 followed by a dose of 1000 mg weekly for 7 weeks and then every 4 weeks for 16 weeks	ORR was 75.3% and only one patient had a CR; Median OS was 20.9 months and PFS was 16.3 months	Diarrhea, pyrexia, neutropenia and fatigue were the most common reported AEs; Grade ≥3 AEs happened in 91% of patients and included neutropenia, diarrhea and pneumonia; 39% of patients discontinued treatment due to AEs	[106]
I	AML	VEGFR; FGFR; PDGFR	Nintedanib twice daily in dosages of 100 mg, 200 mg or 300 mg in a 28-day cycle	Low-dose cytarabine from days 1 to 10 at 20 mg twice daily	2 out of 12 patients had an objective response of CR and CRi; Median OS was 234 days	No DLTs were reported during dose escalation; Most common AEs associated with treatment were gastrointestinal and treatment-related grade ≥3 AEs were reported in 4 patients; A total of 12 serious AEs were reported, the most common being neutropenic fever	[107]
I	AML; CMML	AURK A; Multi-kinase activity	ENMD-2076 administered in daily escalating doses of 225, 275, 325 or 375 mg	NR	ORR was 25% with best response being CRi; Median number of cycles to best response was 2 and median DOR was 4.8 months; 48% of patients discontinued treatment due to disease progression	96% of patients reported treatment related AEs; Some of the most common nonhematologic toxicities were fatigue, diarrhea and hypertension; DLTs were observed in all dose levels except 225 mg/day	[108]

AML: Acute myeloid leukemia; B-NHL: B-cell non-Hodgkin’s lymphoma; CLL: Chronic lymphoid leukemia; CMML: Chronic myelomonocytic leukemia; DLBCL: Diffuse large B-cell lymphoma; FL: Follicular lymphoma; HCL: Hairy cell leukemia; HL: Hodgkin’s lymphoma; MCL: Mantle cell lymphoma; MDS: Myelodysplastic syndrome; MM: Multiple myeloma; MZL: Marginal zone lymphoma; SLL: Small lymphocytic lymphoma; T-NHL: T-cell non-Hodgkin’s lymphoma; G-CSF: Granulocyte colony stimulating factor; IV: Intravenous; OME: Omacetaxine Mepesuccinate; CBR: Clinical benefit rate; CR: Complete remission; CRc: Composite complete remission; CRi: Complete remission with incomplete hematologic recovery; DCR: Disease control rate; DOR: Duration of response; ORR: Overall response rate; OS: Overall Survival; PD: Progressive disease; PFS: Progression free survival; PR: Partial remission; R/R: Relapsed/Refractory; SD: Stable Disease; AE: Adverse event; DLT: Dose limiting toxicity; MTD: Maximum tolerated dose; NR: Not reported.

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
