# Peer review of "Kinase Inhibition in Relapsed/Refractory Leukemia and Lymphoma Settings: Recent Prospects into Clinical Investigations"

_pharmaceutics, 2021, doi:10.3390/pharmaceutics13101604_

Round 1

Reviewer 1 Report

The manuscript by Machado et al. presents a review concerning the use of protein kinase inhibitors (PKIs) treatment in the oncological practice. Research in this direction have been growing in the recent years. Particularly, PKIs can be effectively used in addition to conventional therapy of leukemias and lymphomas. The authors have undertaken an analysis of the literature data in this field.

The manuscript contains a section that provides a historical background of conventional leukemia treatment since this disease was first described in 1827. The problem of drug resistance is addressed and discussed by the authors. The resistance mechanisms in leukemias and lymphomas are described with references to the pertinent literature. In subsequent sections of the manuscript, a role of protein kinases for cancer disease states is highlighted along with the importance of PKIs for leukemia treatment. The most important pharmaceuticals based on PKIs are reviewed, their advantages and disadvantages (e.g. emerging drug resistance) are critically discussed, and the PKI classification into different types is described.

Quite a comprehensive list of recent clinical trials involving PKIs on relapsed/refractory hematological malignancies is given in a convenient tabular form (Table 1 in the manuscript). Along with clinical outcomes, the table contains description of adverse events observed in each of the trials. Analysis of the presented data is given in the manuscript. Applications of PKIs on AML and myeloproliferative disorders, as well as on AML and myeloproliferative disorders, are discussed by the authors. In some important cases, structural peculiarities of enzyme-inhibitor interactions available in the literature are presented. A very important section, from the pharmacologist's point of view, is devoted to toxicity profiles of PKIs.

The paper is well-written and will be interesting to specialists in pharmacology, medicine, and drug design. The manuscript contains illustrations (figures) useful for drug resistance discussion and for undersatnding mechanisms of PKI action. Literature references are up-to-date and relevant. The most important publications from the previous decades are also cited. The list of references does not contain unacceptable self-citation. However, figures 1 and 2 are of insufficient quality, and their resolution should be enhanced.

I recommend acceptance of the manuscript for publication after improvement of the figures.

Author Response

Dear reviewer, my co-authors and I would like to thank you for the suggestions made during this high-quality review and then we present the answer to the question.

We inform that with the reviews and suggestions, we were able to improve the idea presented by our work and we appreciate the opportunity. We hope this review has left the article suitable for publication in this high-impact journal and respect in the area.

Kind Regards.

Response to reviewer 1

I recommend acceptance of the manuscript for publication after improvement of the figures.

R = Figures 1 and 2 had their resolution improved and a new figure was added for a better visualization of the main kinases discussed in this paper.

Reviewer 2 Report

In the manuscript, the authors focus on kinase inhibition for the treatment of relapsed or refractory leukemia and lymphoma by presenting the latest clinical trial perspectives. Congrats on the idea, but the way the topic is presented is monotonous and unattractive to the reader. The manuscript needs several corrections.

  1. What period is the literature review from?
  2. What do the authors mean when they write "tumor genetic diagnosis" (line 37) concerning acute and chronic myeloid leukemia (line 33)?
  3. It would be good to add information on the effects of specific drug groups.
  4. A graphical representation of the targets of each drug group in the cell would be desirable.

Author Response

Dear reviewer, my co-authors and I would like to thank you for the suggestions made during this high-quality review and then we present the answers to the questions.

We inform that with the reviews and suggestions, we were able to improve the idea presented by our work and we appreciate the opportunity. We hope this review has left the article suitable for publication in this high-impact journal and respect in the area.

Kind Regards.

Response to reviewer 2

  1. What period is the literature review from?

R = This review encompasses studies from the past 5 years as it is described in text at the second paragraph of topic 3 and in the title of Table I.

  1. What do the authors mean when they write "tumor genetic diagnosis" (line 37) concerning acute and chronic myeloid leukemia (line 33)?

R = The term was used to reference the characterization of a tumor genetic profile in the clinical practice. However, the highlighted term was changed to “tumors genetic profiles” in order for a clearer understanding of the topic.

  1. It would be good to add information on the effects of specific drug groups.

R = Topics 4, 5 and 6 are dedicated to the discussion of Table I and the most prevalent drug groups have their molecular targets and mechanisms of action described as well as patients’ clinical outcomes and the expected toxicity profiles of drug groups and individual drugs.

  1. A graphical representation of the targets of each drug group in the cell would be desirable.

R = A new figure has been developed in order for a better visualization of the main kinases discussed in this paper (Figure 3).
